

# Very low-frequency IEPE accelerometer calibration and application to a wind energy structure

Clemens Jonscher[1], Benedikt Hofmeister[1], Tanja Grießmann[1], and Raimund Rolfes[1]

[1]Leibniz University Hannover / ForWind, Institute of Structural Analysis, Appelstraße 9A, 30167 Hannover, Germany

**Correspondence:** Clemens Jonscher (c.jonscher@isd.uni-hannover.de)

**Abstract.** In this work, we present an experimental setup for very low-frequency calibration measurements of low-noise Integrated Electronics Piezo Electric (IEPE) accelerometers and a customised signal conditioner design for using IEPE sensor down to $0.05\,\mathrm{Hz}$. AC-response IEPE accelerometer and signal conditioners have amplitude and phase deviations at low frequencies. As the standard calibration procedure in the low-frequency range is technically challenging, IEPE accelerometers with standard signal conditioners are usually used in frequency ranges above $1\,\mathrm{Hz}$. Vibrations on structures with low eigenfrequencies like wind turbines are thus often monitored using DC-coupled micro-electro-mechanical systems (MEMS) capacitive accelerometers. This sensor type suffers from higher noise levels compared to IEPE sensors. To apply IEPE sensors instead of MEMS sensors, in this work the calibration of the entire measurement chain of three different IEPE sensors with the customised signal conditioner is performed with a low-frequency centrifuge. The IEPE sensors are modelled using IIR filters to apply the calibration to time-domain measurement data of a wind turbine support structure. This procedure enables an amplitude and phase-accurate vibration analysis with IEPE sensors in the low-frequency range down to $0.05\,\mathrm{Hz}$.

## 1 Introduction

In recent years, the expansion of offshore wind energy has been driven forward with ever larger wind turbines. This leads to ever smaller natural frequencies of wind turbine support structures. Waves in the low-frequency range down to $0.05\,\mathrm{Hz}$ also have an impact on these structures. For instance, Penner et al. (2020) observed that the highest forces and displacements occur in the frequency range between $0.05$ and $0.2\,\mathrm{Hz}$ when monitoring a suction bucket offshore foundation. Therefore, low-frequency structural dynamics should be considered when monitoring such structures. Structural health monitoring (SHM) based on dynamic measurements relies on measurement data from a sensor network installed on the structure to be monitored. For a reliable monitoring of support structures of offshore wind turbines, the measurement chain should be designed for the low-frequency range. For onshore settings, the displacement for this frequency range can for instance be measured using photogrammetry (Ozbek et al., 2010). However, optical measurement systems require fixed reference points, which is generally not available for offshore installations. Furthermore, the resolution of the camera limits the obtainable signal-to-noise ratio (SNR). Strain gauges can also be used to monitor low frequency vibration. However, field experiences show that strain sensors are less reliable than accelerometers for long-term offshore applications (Maes et al., 2016). Therefore, various virtual sensing concepts have been developed to estimate dynamic strains at fatigue critical locations using accelerometers (e.g. Tarpø et al.,



2020). In the low frequency range, DC-coupled Micro-electro-mechanical systems (MEMS) capacitive accelerometers are usually applied, because these sensors have a linear transfer behaviour in the low frequency range. A relatively high noise level is a disadvantage of this sensor type. This limits the range of application of MEMS sensors, since a high SNR is an important prerequisite for reliable displacement and strain estimation using accelerometers. In addition, a high SNR also leads to better identification of modal parameters (Au, 2014).

Regarding low-noise accelerometers, the Integrated Electronics Piezo Electric (IEPE) sensor type is the industry standard. This type of sensor is a Piezo Electric (PE) sensor with a preamplifier integrated into the sensor casing. In contrast to conventional PE sensors, this leads to a low output impedance, which results in a significantly improved noise behaviour (Levinzon, 2005). The integrated preamplifier requires a constant current source. To connect the sensor with standard analogue digital converters (ADC) a high-pass filter is integrated into the supply. The sensor supply consisting of the current source and the filter is also called IEPE signal conditioner. Due to the measurement principle, IEPE sensors are AC-response sensors. This sensor class cannot measure constant acceleration, leading to a frequency-dependent transfer behaviour. Thus, low-noise IEPE sensors are typically used in the frequency range above $1\,\mathrm{Hz}$. In order to correct measurement errors in the frequency range below $1\,\mathrm{Hz}$, the transfer behaviour should be represented using a filter model. The simplest model of a measuring chain with an IEPE sensor consists of two cascaded first order high-passes (D'Emilia et al., 2019). To determine the filter coefficients, it is necessary to calibrate the sensor and the signal conditioners below $1\,\mathrm{Hz}$.

The transfer behaviour of an IEPE signal conditioner can be analysed using a frequency generator and an IEPE simulator (Ripper et al., 2014). Klaus et al. (2015) calibrated different IEPE signal conditioners in the frequency range from $0.1\,\mathrm{Hz}$ to $100\,\mathrm{kHz}$ using a sinusoidal excitation. It was shown that the different designs of the built-in high-pass filters lead to large deviations in the frequency range below $3\,\mathrm{Hz}$.

The calibration of acceleration sensors is regulated in the ISO 16063 "Methods for the calibration of vibration and shock transducers" series of standards. ISO 16063-21 regulates the calibration using a reference sensor in the frequency range from $0.4\,\mathrm{Hz}$ to $10\,\mathrm{kHz}$. In the calibration procedure, an acceleration sensor is excited using a electrodynamic shaker. In addition, the excitation is measured using a calibrated reference acceleration sensor. In the low-frequency range, long-stroke shakers are used for the calibration, so that sufficient displacement is achieved.

To be able to calibrate frequencies down to $0.002\,\mathrm{Hz}$, He et al. (2014) developed a special long-stroke shaker with a stroke of one meter. However, the amplitudes in the low frequency range are still very low. To achieve higher amplitudes at low frequency, the sensor can also be rotated in the earth's gravity field (Dosch, 2007). This results in acceleration amplitudes of $\pm 1\,\mathrm{g}$ independently from the rotation frequency. Seismic sensors can have a measuring range smaller than $2\,\mathrm{g}$. The acceleration amplitude can therefore be adjusted by tilting the centrifuge. For example, Olivares et al. (2009) describe a tilted non-motorised centrifuge which is used to calibrate a gyroscope.

In addition to the frequency response, the spectral noise level is also an important parameter for evaluating a measurement chain. A widely used method to measure the noise level is the Huddle Test (Holcomb, 1989). In this test, several sensors are measured simultaneously, while the external accelerations of all sensors has to be the same. This is achieved by mounting the sensors to a stiff plate and aligning them in the same direction. When using two sensors, it is assumed that both sensors have



the same noise level. For three sensors, the three channel test is recommend to determine the noise level of each individual sensor (Sleeman et al., 2006).

In this work, we present an approach for the design of very low-frequency measurement chains for low-noise IEPE accelerometers. This measurement chain can be used in different applications, such as vibration-based SHM in heterogenous sensor setups or load monitoring in offshore wind turbines support structures. Our approach is to use a custom IEPE signal conditioner with a low cut-off frequency to achieve a higher SNR compared to standard signal conditioner. To determine the transfer behaviour, we apply a motorised centrifuge to perform a low frequency calibration between $0.027\,\mathrm{Hz}$ and $1\,\mathrm{Hz}$. The limits of this frequency bands are determined by the technical limitations of the centrifuge. Using this approach, constant acceleration up to $\pm 1\,\mathrm{g}$ is possible in the low-frequency range with a cost-effective experimental setup. We calibrate three different IEPE sensors to study differences in their transfer behaviour. To apply the calibration results to measurement data, a filter model is identified for each sensor. This is used to investigate the physical noise level with and without calibration. Finally, the filter models are applied to measurements of other calibration procedures as well as to measurements of tower vibrations of a wind turbine in order to demonstrate calibration of time-domain measurement data down to $0.05\,\mathrm{Hz}$.

## 2 Theory

In this section, we present the theoretical foundation of the proposed calibrated measuring chain. First, we give a summary of IEPE sensor technology. Then we introduce the theory of calibration of accelerometers using a centrifuge. For data evaluation and further processing, the Vold-Kalman filter and other filter theories are presented. Finally, the numerical methods for the identification of the calibration filter coefficients are introduced.

### 2.1 IEPE Sensors

Piezoelectric sensors have been used in vibration analysis for frequency ranges above $1\,\mathrm{Hz}$ for a long time. In the first generation of such devices, the piezo element is directly connected to the measurement line. This results in a high-impedance setup with very low current in the measurement lines. Due to cable microphonics and the susceptibility to stray fields, noise and hum issues, especially in setups with long measurement cabling, the signal quality deteriorates. This can be improved to a limited extent by using very expensive, highly shielded cables with low microphonic interference. In an industrial atmosphere, however, mechanical vibrations and electromagnetic interference have to be expected.

The current generation of piezoelectric sensors are IEPE devices. The key difference to the previous generation is a preamplifier, which is integrated into the sensor casing. For the measurement system, the sensor thus becomes a low-impedance load, which leads to an improved noise characteristic (Levinzon, 2005). The IEPE sensor is a two-terminal design, which is realised by employing a Field Effect Transistor (FET) with the gate connected to the piezo crystal. The preamplifier is powered by an IEPE signal conditioner, which provides a constant current and a bias voltage of around $10\,\mathrm{V}$. The IEPE sensor typically has a measuring range of $\pm 5\,\mathrm{V}$. The sum of the bias voltage and and the measuring range delivers an output voltage of $5\,\mathrm{V}$ to $15\,\mathrm{V}$.





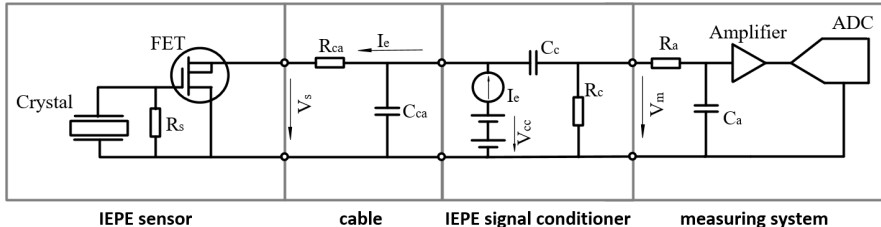

**Figure 1.** Schematic diagram of a measurement chain of an IEPE sensor.

To interface with standard Analogue-Digital Converters (ADC), a coupling capacitor $C_C$ is introduced, as shown in Figure 1. This coupling capacitor separates the constant excitation current $I_e$, and thus the bias voltage, from the ADC. For zero acceleration, the voltage at the ADC input is thus zero. A large resistance $R_C$ is placed across the ADC, which defines the output impedance and enables charging of the coupling capacitor. This decoupling circuit results in a first order high-pass filter behaviour with a cutoff frequency of

$$f_c = \frac{1}{2\pi R_C C_C}. \tag{1}$$

The time constant

$$\tau = R_C C_C \tag{2}$$

is defined by the time required for the high pass to decay to 67% of the output value of a step response. After some time, the measured signal thus becomes zero for constant accelerations. The same effect takes place inside the sensor as well, since the piezo crystal discharges due to leakage current. This is the reason, why this type of accelerometer cannot be used to measure constant accelerations. Inside the sensor, a resistor $R_S$ is placed parallel to the piezo element to limit the settling time and avoid temperature drifting as shown in Figure 1. This further elevates the cutoff frequency. For low frequency applications, the resistor $R_S$ needs to have a large value, which leads to long settling times of about several minutes. To optimise the transfer behaviour of the sensor, further electronics are installed in the sensor by the manufacturer. Seismic grade sensors can attain cutoff frequencies below $0.1\,\mathrm{Hz}$, coupled with very low noise and high sensitivity (Levinzon, 2012).

To exploit the full range of IEPE sensors, the cutoff frequency of the IEPE signal conditioner must be lower than that of the sensor itself. Off-the-rack measurement systems with integrated IEPE signal conditioner usually have cutoff frequencies around $0.4\,\mathrm{Hz}$. This is due to space restrictions in the casing, since film capacitors with the required capacitance rating have case dimensions of several centimetres.

Discrete IEPE signal conditioner units typically achieve a cutoff frequency of $0.1\,\mathrm{Hz}$, consequently they have larger case dimensions. Regarding the amplitude response, this results in an acceptably low amplitude loss. However, the phase response is still affected. Signal conditioners with even lower cutoff frequencies lead to longer settling times, which is undesirable for most applications. However, some manufacturers offer special versions with a long settling times for low frequency applications.





In addition to the sensor and the signal conditioner, a measuring channel also consists of the cable and the measuring system. Since these components have a linear response in the low frequency range, they play a minor role and are not considered further.

## 2.2 Calibration of the measurement chain

A measured signal $y$ generally consists of a deterministic signal component $s$ and a stochastic noise component $n$

$$y[k] = s[k] + n[k], \tag{3}$$

where the index k is the time variable of the time-discrete signal normalised to its sampling rate. The deterministic part corresponds to the physical quantity to be measured. The stochastic part is attributed to the noise of the measuring chain. It consists of a frequency-independent component (white noise) and a frequency-dependent component (1/f noise or pink noise). In the low-frequency range, 1/f noise is the decisive component. The Huddle test is employed to investigate the incoherent

noise of the accelerometer measurement chain. In this test, at least two identical sensors are placed as close as possible to each other. By means of the coherence function $\gamma_{1,2}$ among the two sensors, the auto power density spectrum of both signals $S_{1,1}$ and $S_{2,2}$ can be separated into the signal component $S_{s,s}$

$$S_{s,s}[f] = \gamma_{1,2}[f] \sqrt{S_{1,1}[f] S_{2,2}[f]} \tag{4}$$

and noise component $S_{n,n}$

$$S_{n,n}[f] = (1 - \gamma_{1,2}[f]) \sqrt{S_{1,1}[f] S_{2,2}[f]}, \tag{5}$$

depending on the frequency $f$ (Brincker and Larsen, 2007). For the calibration of the deterministic signal component, it is assumed that the entire measurement chain is a linear time-invariant system. Hence, the frequency response is not dependent on the time or the amplitude of the input (e.g. Klaus et al., 2015). Therefore, the transfer between signal input $x(z)$ and output $y(z)$ can be described with the time-invariant transfer function $H(z)$

$$y(z) = H(z)\, x(z), \tag{6}$$

where $z$ is the discrete frequency obtained from the $z$-transform. The aim of the calibration is to determine the transfer function. Various excitation signals can be used for calibration, the most common being a mono-frequent sinusoidal signal

$$x(t) = A \sin(\Omega t + \varphi), \tag{7}$$

with the amplitude $A$, the angular frequency $\Omega$ and the phase shift $\varphi$. In order to calibrate the low-frequency range, a long

measuring time with a mono-frequent sinusoidal signal is necessary. Therefore, a multi-sinus excitation can be used to reduce the length of the time series required for the calibration (Bruns and Volkers, 2018)

$$x(t) = \sum_{k=1}^{n} A_k \sin(\Omega_k t + \varphi_k). \tag{8}$$



For the calibration of an IEPE signal conditioner, there is already an established procedure (e.g. Ripper et al., 2014; Klaus et al., 2015). Following this approach, an excitation signal is generated by a signal generator. The signal type used for this procedure

is arbitrary and only limited by the type of waveforms the signal generator can generate. Due to the electrical impedance and bias voltage mismatch, a signal generator cannot be directly connected to the IEPE signal conditioner. Therefore, an IEPE simulator is used as an impedance converter, which is connected between the signal generator and the IEPE signal conditioner. This calibration setup is shown in Figure 2. The standard ISO 16063-21 for the calibration of acceleration sensors proposes

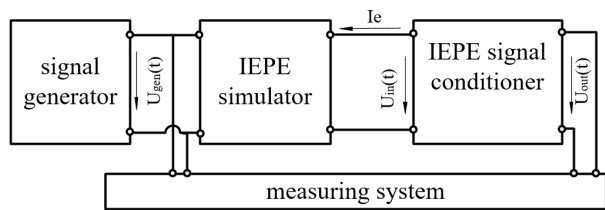

**Figure 2.** Measurement Setup of an IEPE signal conditioner, inspired by Klaus et al. (2015).

a long-stroke shaker for the calibration of sensors in the low-frequency range. Using a shaker, the obtainable acceleration

amplitude is very low in the low-frequency range due to the physical relationship between displacement $u$ and acceleration signal $a$ of a harmonic signal

$$a[f] = (2\pi f)^2 u[f],\tag{9}$$

which makes shaker-based calibration below 1 Hz technically challenging and expensive.

In order to obtain a frequency-independent acceleration for the calibration, a possibility is to employ the earth's gravity

field and an inclined plane of rotation such as Olivares et al. (2009) proposed to calibrate a gyroscope. The rotation of the centrifuge shown in Figure 3 leads to a tilting motion relative to the gravitational acceleration $g$. The acceleration resulting from this motion depends on the tilting angle $\phi$ and the angular frequency $\Omega$. If the sensors measured in the tangential direction of the rotational motion, one revolution of the centrifuge translates to one oscillation period for the sensor. In addition to the gravitational acceleration $\mathbf{a}_{\mathrm{grav}}$, the centripetal acceleration $\mathbf{a}_{\mathrm{cent}}$ acts on the sensor as well. There are additional influences on

acceleration, such as higher harmonics of the centrifuge motor and measuring uncertainly, which are summarised in the therm $\mathbf{a}_e$. The acceleration acting on the sensor is thus

$$\mathbf{a} = \mathbf{a}_{\mathrm{cent}} + \mathbf{a}_{\mathrm{grav}} + \mathbf{a}_e.\tag{10}$$

Acceleration due to gravity measured at the sensor depends on the tilting angle $\theta$ of the centrifuge, the angular velocity $\Omega t$ and the gravitational acceleration $g$

$$\mathbf{a}_{\mathrm{grav}} = \begin{bmatrix} g\sin\theta\cos\Omega t \\ g\sin\theta\sin\Omega t \\ g\cos\theta \end{bmatrix},\tag{11}$$



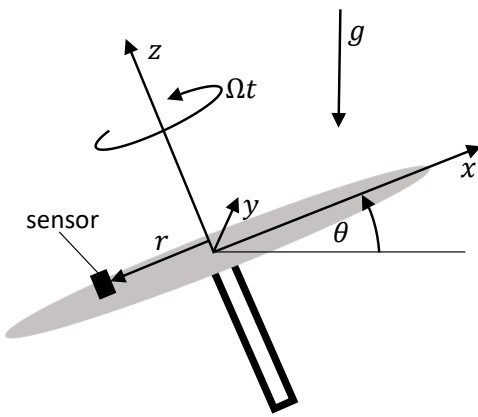

**Figure 3.** Experimental setup

The centripetal acceleration is determined by the distance from the rotation axis and the angular velocity. For a constant angular velocity, the centripetal acceleration is

$$
\mathbf{a}_{\text{cent}} = \begin{bmatrix} r_x \Omega^2 \\ r_y \Omega^2 \\ 0 \end{bmatrix}, \tag{12}
$$

where $r_x$ and $r_y$ denote the position of the sensor relative to the axis of rotation. The centripetal acceleration acts in the radial
direction, perpendicular to the axis of rotation. To avoid distortion of the measured signal the centripetal acceleration should be as low as possible. Also the centripetal acceleration is constant and thus cannot be measured with IEPE sensors, it can, due to the transverse sensitivity of the sensor, negatively impact the measurement result. In case of DC capable MEMS sensors, the centripetal acceleration can easily be removed by digital filtering, since it is constant.

### 2.3    Vold-Kalman Filter

The measured acceleration data obtained at the centrifuge is contaminated with noise and higher harmonic frequency components and in case of MEMS sensors also with centripetal acceleration. Digital filtering is thus required for an accurate determination of amplitude and phase. To avoid phase shifts, we use the second generation Vold-Kalman filter (Vold and Leuridan, 1993). This filter is a time domain decomposition method for order-tracking. For a given phase signal, the filter extracts only the harmonic component from the measured signal. The method is based on two equations for each time step, which are



minimised in a system of equations over the entire data length. The data equation ensures that the signal components $a_e[k]$ that do not originate from the harmonic component are minimised

$$a_e[k] = a[k] - A[k]e^{j\omega[k]}, \tag{13}$$

where $a[k]$ is the measured signal, $A[k]$ is the instantaneous amplitude and $\omega[k]$ is the given phase signal. The second equation is the structural equation. This equation leads to a smooth amplitude trend by keeping the change in amplitude as low as possible over several time steps $k$. Thus, it acts as a low pass filter for the amplitude. Therefore, abrupt amplitude changes in the measurement data lead to transient oscillations at the filter output. The structural equation depends on the filter order. For a first-order filter, the equation is

$$\eta[k] = A[k] - A[k+1], \tag{14}$$

where $\eta[k]$ is the change of the amplitude. The entire system of equations is solved using a least squares algorithm. The filter property is changed by a weighting factor between the structural equation and the data equation. This weighting factor can be calculated from the so-called filter bandwidth $B$ (Tuma, 2005). In this work, we calculate the filter bandwidth

$$B = \kappa\Omega, \tag{15}$$

with the bandwidth factor $\kappa$. The bandwidth thus depends on the excitation frequency $\Omega$. The more accurate the phase signal and more constant the amplitude, the smaller the factor $\kappa$ can be selected. A small bandwidth leads to a more precise determination of amplitude and phase. However, the settling time of the filter increases with decreasing bandwidth as described by Tuma (2005) and Herlufsen et al. (1999). Due to the non-causal filter characteristics, the Vold-Kalman filter cannot be applied in real-time and has to be used as a post-processing method.

### 2.4 Filter model of the transfer behaviour of the measurement chain

The IEPE signal conditioner and the IEPE sensor act as high-pass filters (D'Emilia et al., 2019). The discrete transfer function of a first order high-pass filter can be expressed as

$$H_{\mathrm{HP}}(z) = \frac{B(z)}{A(z)} = \frac{-\alpha + \alpha z^{-1}}{\alpha + z^{-1}} \text{ and } \alpha = e^{-2\pi f_c T_s}, \tag{16}$$

where $z$ is the discrete frequency obtained from the $z$-transform and $T_s$ is the period duration of the sampling frequency. The cutoff frequency $f_c$ of an RC filter can be calculated according to Equation 1. The transfer behaviour of an exemplary high-pass filter is shown in the Bode diagram in Figure 4. By multiplication of the filters in the $z$-domain, several filters can be combined

$$H(z) = \prod_{i}^{N} H_{\mathrm{HP},i}(z). \tag{17}$$





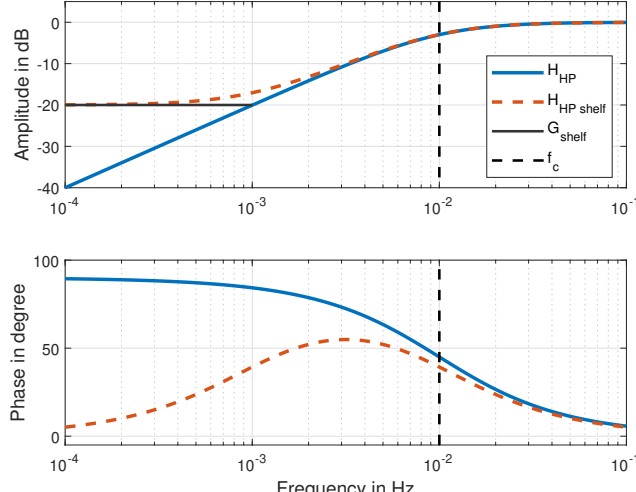

**Figure 4.** Bode diagram of a high-pass of first order with a cutoff frequency of $0.01\,\mathrm{Hz}$ and shelving high-pass with $G_{\mathrm{shelf}}$ of $-20\,\mathrm{dB}$

To calibrate the measurement data, the filter representing the measurement chain has to be inverted. The transfer function can be inverted by exchanging the numerator and denominator coefficients

$$H^{-1}(z) = \frac{1}{H(z)} = \frac{A(z)}{B(z)}. \tag{18}$$

The high-pass filter characteristic of the IEPE measurement chain removes the mean acceleration from the signal so that it cannot be reconstructed in the calibration. In addition, 1/f noise enter the signal from the measurement chain, which can lead to drift when the high-pass filter is inverted. This is due to the pole of the inverted high-pass at 0 Hz, which makes the filter semi-stable. In order to enable calibration in the low-frequency range, a shelving high-pass filter can be used. A shelf gain $G_{\mathrm{shelf}}$ is introduced to limit the amplitude in the low-frequency range. The transfer function for a shelved high-pass filter is

$$H_{shelf}(z) = (1 - G_{\mathrm{shelf}})H(z) + G_{\mathrm{shelf}}. \tag{19}$$

Using this shelved filter, low-frequency components are limited in amplitude when the filter is inverted. This prevents drifting and thus leads to valid signals when the calibration is conducted. The Bode diagram of a high-pass filter with and without shelf is shown in Figure 4. However, the shelf filter introduces a phase error below the cutoff frequency. Therefore, the phase behaviour should be carefully considered when selecting the shelf gain.

**2.5   Identification of filter parameters**

In order to identify filter coefficients for the measured transfer functions, a parameter identification is required. This is accomplished using a numerical optimisation method. As the objective function $\epsilon$ we use a weighted Euclidean distance between the





measured and modelled complex transfer function

$$\min_{f_c} \epsilon(f_c) =$$

$$\min_{f_c} \sqrt{\sum_{i=1}^{n} \left( \frac{1}{|H_{\text{meas},i}|} |H_{\text{model},i}(f_c) - H_{\text{meas},i}| \right)^2}, \tag{20}$$

where $H_{\text{meas}}$ is the measured and $H_{\text{model}}$ is the modelled transfer function. A weighting based on the measured transfer function is necessary because the magnitude of the transfer function $H$ of a high-pass filter below the cutoff frequency approaches zero quickly. This is shown in Figure 5 a). In order to weight each measured point equally in the curve, a normalisation with the inverse of the absolute measured transfer function is performed as shown in Figure 5 b). This plot demonstrates, that the real part dominates above the cut-off frequency and the imaginary part dominates below it. Equation 20 can be solved for the cutoff

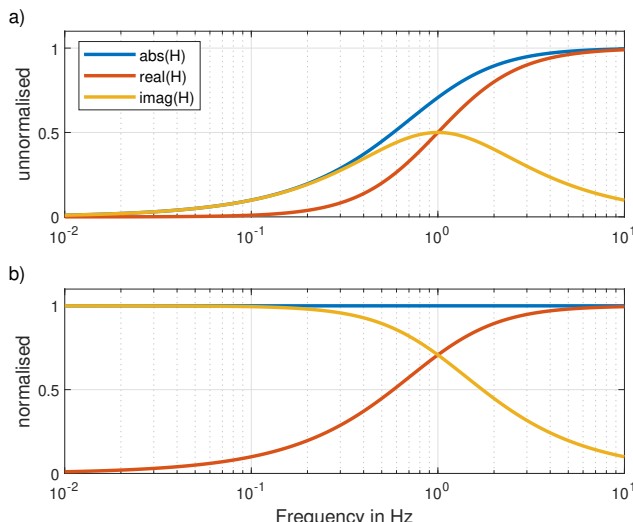

**Figure 5.** Absolute value, real and imaginary part of the transfer function of the high-pass shown in Figure 4: a) unnormalised, b) normalised

frequency using a global optimisation algorithm, such as the global pattern search algorithm (Hofmeister et al., 2019).

To calibrate the measurement data, the identified filter model is inversely applied to the measurement data. The well-known effect of the drift of the measured data after time integration can also be observed when applying the inverse filter. By employing shelving high-pass filters, the increase in amplitude in the low frequency range can be limited, thus preventing drift. Due to the phase error introduced by the shelf, there is a conflict of goals between phase fidelity and amplitude limiting, as illustrated in

Figure 4. For post-processing calibration, the amplitude in the low-frequency range can be reduced without changing the phase by applying a high-pass filter forwards and backwards in time. To design this high-pass filter, the design variable can be the maximum gain $G_{max}$ of the measurement signal through the filter sequence. Using the objective function, the corresponding cut-off frequency of the high-pass filter can be determined by means of numerical optimisation. The identification of the cutoff





frequency for a given order of the high-pass filter can be done by means of the target function

$$\min_{f_c} \epsilon = \min_{f_c} \left| \max \left( \left| H_{\text{model}}^{-1} H_{\text{HP}}(f_c) \right| \right) - G_{max} \right|, \tag{21}$$

where $H_{\text{HP}}(z)$ is the transfer function of the high-pass.

## 3 IEPE signal conditioner circuit

For a precise investigation of the sensor behaviour in the low-frequency range, an IEPE signal conditioner with a low cutoff frequency and low noise is required. Therefore, we propose a custom IEPE signal conditioner circuitry which fulfils these design criteria. The circuit diagram is shown in Figure 1. As a current source, the LT3092 integrated circuit is used. The coupling capacitor $C_C$ is a foil type with a low dissipation factor and with a capacitance of $47\,\mu\text{F}$. The resistor $R_C$ has an electrical resistance of $330\,\text{k}\Omega$. According to Equation 1, the cutoff frequency of this signal conditioner design is $0.0103\,\text{Hz}$. Figure 4 shows the theoretical transfer behaviour of the resulting high-pass filter.

To enable long distance cabling in adverse electromagnetic conditions, we implement a shielding concept. We use twisted pair cabling for the signal ground and sense wires to protect against magnetic fields. A common copper mesh shields against interference from electric fields. The standard connector on industrial grade IEPE accelerometers is of the type MILC-5015, which enables a full enclosure of the signal wires inside the metallic shield. For the signal conditioner, we use cheaper XLR connectors instead of MIL-type connectors. This type of connector also fully encapsulates the signal wires and mechanically ensures inverse polarity protection. The sensor cables are thus designed to convert from XLR male to MILC-5015 female connectors. The housing and electrical circuit board of the signal conditioner are shown in Figure 6.

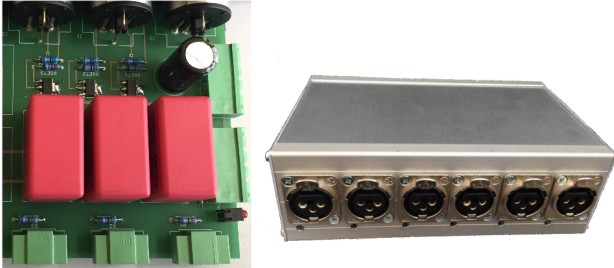

**Figure 6.** Housing and circuit board of custom IEPE signal conditioner

### 3.1 Calibration of the IEPE signal conditioner

In order to check the functionality of the custom IEPE signal conditioner and to obtain the exact transfer function, we carry out the calibration according to Ripper et al. (2014). For comparison, the integrated signal conditioner of the measuring system is additionally calibrated. According to the data sheet, it has a cutoff frequency of $0.34\,\text{Hz}$. The measurement setup is shown in Figure 2. We employ an IEPE simulator with a flat low-frequency response down to DC. To speed up the calibration





measurement, a multi-sine signal is fed to the IEPE simulator using a signal generator. Therefore, a signal with one fundamental and seven higher harmonics is applied with an amplitude of 0.5V each. After a settling time of two minutes, 18 periods of the fundamental oscillation or at least five minutes measuring time are used for the calibration.

The data analysis is carried out using the second generation first order Vold-Kalman filter described in Section 2.3. We set
the bandwidth factor introduced in Equation 15 to $\kappa = 0.001$. In the evaluation, the first and last six periods of the fundamental oscillation or at least 120 seconds are not used due to the transient response of the Vold-Kalman filter. The phase signal required for filtering is calculated from the known frequencies of the signal generator.

The calibration of the signal conditioners results in an amplitude dispersion of less than $0.02\%$ and a phase scatter below $\pm0.01°$. Besides the Vold-Kalman filter, the signal generator, the IEPE simulator and the measuring system are the contributors
to this measurement uncertainty. Klaus et al. (2015) estimate the expanded uncertainty of this calibration method in the per mille range, which is consistent with our results.

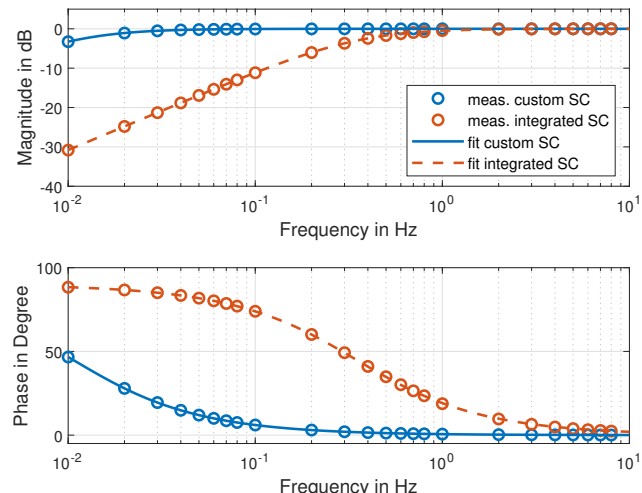

**Figure 7.** Results and filter model of the signal conditioner (SC) calibration: custom signal conditioner: $f_c$=0.0106 Hz, integrated signal conditioner: $f_c$=0.3474 Hz

The frequency response of both signal conditioners are shown in Figure 7. The Bode diagrams resemble the transfer behaviour of first order high-pass filters as described in Equation 16. Model identification using the identification approach proposed in Section 2.5 provides the cutoff frequencies of the signal conditioners. The cutoff frequency of the custom signal
conditioner is $0.0106$ Hz and for the integrated signal conditioner it is $0.3474$ Hz. The results of the model identification are also shown in Figure 7. The measurement data of both sensor supplies fit very well with the model. Both cutoff frequencies of the power supplies are higher than the theoretical or manufacturer's specifications. One cause could be the input impedance of the measuring device, which is connected in parallel to the signal conditioner output and thus reduces the effective value of $R_c$.



The determined transfer functions clearly show, that for the application of IEPE sensors in the low-frequency range, the transfer behaviour of the signal conditioner should be examined in detail. If the cutoff frequency of the signal conditioner is in the frequency range to be measured, there is a considerable amplitude error. This can be calibrated, but the SNR will suffer. If the cutoff frequency is lower, the amplitude error is small, but a significant phase error still remains. The phase distortion leads to a group delay, which is imposed onto the signal characteristics.

In the following section, the calibration of the entire measurement chain down to 0.027 Hz is carried out. The custom signal conditioner is used for calibration, as it leads to a higher SNR in the low-frequency range due to the lower cutoff frequency.

## 4    Low-frequency calibration of IEPE accelerometer measuring chains

In addition to the IEPE conditioner, the sensor itself has a high-pass characteristic. Therefore, the sensor should be calibrated for accurate measurement in the low-frequency range as well. We perform the procedure considering the entire measurement
chain on a custom motorised centrifuge with a tilted plane of rotation. The calibration is carried out in the frequency range from 0.027 Hz to 1 Hz. The phase signal required for the application of the Vold-Kalman filter is measured by an optical tachometer. The setup is shown in Figure 8.The inclination angle of the rotation plate is

$$\theta = 2.46°. \tag{22}$$

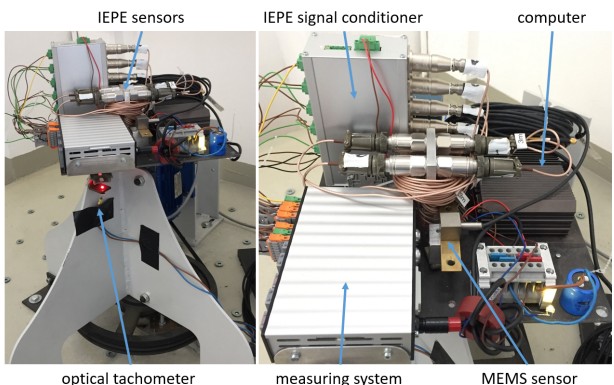

**Figure 8.** Measurement setup for sensor calibration using a centrifuge

According to information by the German Federal Agency for Cartography and Geodesy, the gravitational acceleration is $g = 981262.8$mGal at the site of the centrifuge. According to Equation 11, the expected frequency-independent acceleration in the measurement plane is

$$a = g\sin(\theta)\cos(\Omega t) = 0.421\frac{m}{s^2}\cos(\Omega t). \tag{23}$$





**Table 1.** Sensor characteristics

|  | IEPE A | IEPE B | IEPE C | MEMS |
|---|---|---|---|---|
| Measurement range in g | ± 5 | ± 5 | ± 0.5 | ±2 |
| Sensitivity in $mV/m/s^2$ | 101.937 | 101.937 | 1019.37 | 101.9 |
| Frequency response, ±5% in Hz | - | 0.3 - 4000 | 0.15 - 1000 | 0 - 250 |
| Frequency response, ±3dB in Hz | 0.04 - 6000 | 0.1 - 6000 | 0.05 - 4000 | - |
| transverse sensitivity in % | <5 | <5 | <7 | - |

For the calibration measurement, we use a 24-bit Delta Sigma AD converter with a sampling rate of $2500\,\mathrm{Hz}$ and a measuring
range of $\pm\,10$ V. A digital Bessel filter with a cutoff frequency of $500\,\mathrm{Hz}$ is used to prevent aliasing. The AD converter is
connected to the previously calibrated custom IEPE signal conditioner and mounted on the rotating platform of the centrifuge.
We calibrate three different IEPE sensors to investigate differences in their low-frequency transfer behaviour. The first and
second sensors (IEPE A, B) are low frequency variants of a general purpose sensor. The third sensor IEPE C is a seismic high
sensitivity type. For comparison, a DC capable MEMS sensor is calibrated in addition to the IEPE sensors. The characteristics
of these sensors are listed in Table 1.

The centrifuge is controlled so that each calibration frequency has at least 50 oscillation periods and a minimum measuring
time of 200 seconds. We chose 17 equally spaced calibration frequencies in the frequency range of $0.027\,\mathrm{Hz}$ to $1\,\mathrm{Hz}$ on a
logarithmic scale. The time required to complete the calibration procedure with these parameters is 152 minutes. For the
sensor calibration, we ignore the first and last 12 oscillation rotations for each frequency so that transient oscillations of the
centrifuge and the Vold-Kalman filter do not falsify the resulting frequency response data.

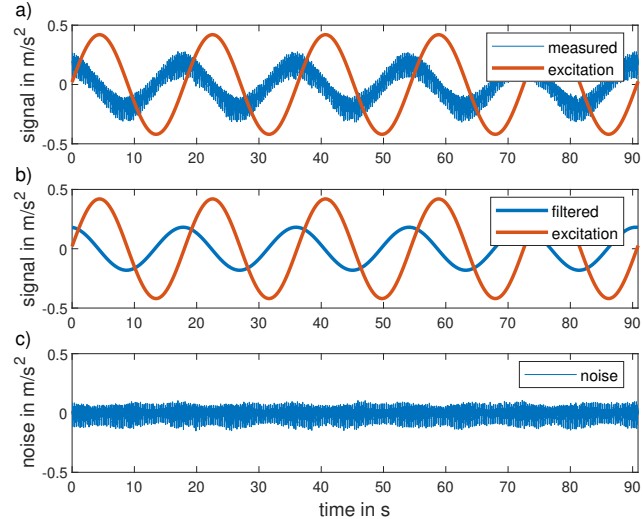

**Figure 9.** Measurement results and filtering of 5 oscillation periods with a frequency of 0.055Hz from IEPE B




The data measured on the centrifuge is contaminated with higher harmonic oscillations and other disturbing effects. Panel a) of Figure 9 shows five periods of the calibration measured with the sensor IEPE B at an excitation frequency of 0.055 Hz. The Vold-Kalman filter is applied to extract only the fundamental oscillation component of the signal. The required phase signal is calculated on the basis of the tachometer signal, which triggers once per revolution of the centrifuge. Figure 10 shows

the influence on the selected bandwidth factor of the Vold-Kalman filter for the IEPE B sensor at an excitation frequency of 0.055 Hz. Panel a) shows the least square error between the Vold-Kalman filter and a bandpass filter (passband 0.0275 Hz-0.0825 Hz) as a function of the bandwidth factor. A bandwidth factor that is too high will result in larger errors. Panel b) and c) show the resulting amplitude and phase between the excitation and measurement. Below a bandwidth factor of 0.003, the amplitude increases briefly. This is probably a numerical effect. Therefore, we set the bandwidth factor of the Vold-Kalman

filter to $\kappa = 0.005$ for the further analysis. Such a low bandwidth factor is possible due to no amplitude changes are to be expected during one excitation frequency.

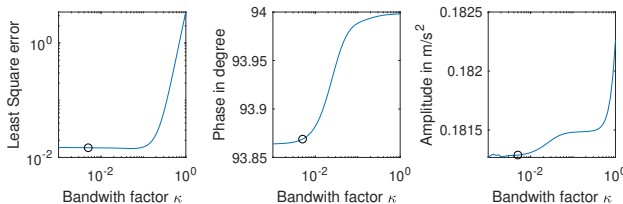

**Figure 10.** Influence of the bandwidth factor on the least square error, amplitude and phase evaluated for the IEPE B sensor at an excitation frequency of 0.055 Hz

The result of the filtering with the Vold-Kalman filter is shown in panel b) of Figure 9. The filter output contains a signal which is phase-shifted and attenuated when compared to the excitation. This error results from the combined frequency response of the sensor and the signal conditioner. Panel c) shows the noise signal resulting from the difference between the

measured signal and the filtered signal.

The calibration results are verified statistically by computing the average values and the minima and maxima from the instantaneous phase and amplitude obtained from the Vold-Kalman filter. Figure 11 shows the relative deviation of the amplitude and the absolute phase deviation of all investigated sensors. The amplitude varies up to 0.4% and the phase deviation is below 0.2°. One reason for the scattering of the phase at higher frequencies is the low quality of the employed phase trigger. With

elevated speed of the centrifuge, the phase signal becomes less accurate, which is reflected in the statistical scatter. Moreover, mechanical warping due to centrifugal forces at higher frequencies can lead to higher deviations.

The results of the calibration are shown in Figure 12. All three IEPE sensors have a typical high-pass characteristic of higher order. A significant difference in amplitude between the three sensors can be observed below 0.2 Hz, while the phase differs significantly below 0.5 Hz. As expected, the MEMS sensor has a linear transfer behaviour in the low-frequency range which

results in a flat amplitude and phase response. It should be noted that the IEPE C was calibrated in another calibration run with slightly different frequencies, which result from friction in the drivetrain of the centrifuge.

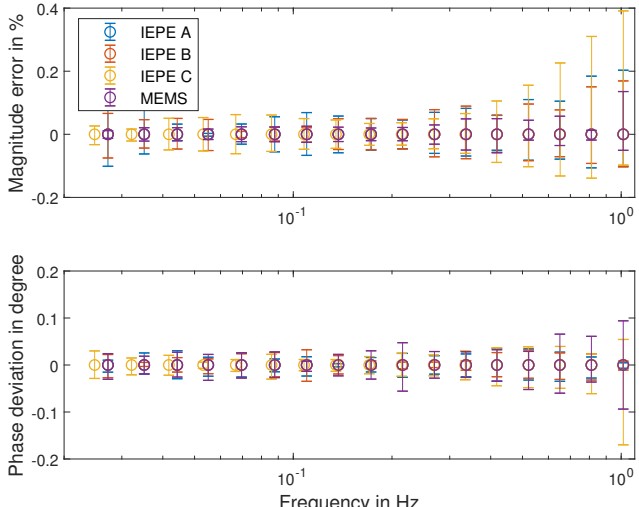

**Figure 11.** Minimal and maximal deviation in the analysis of the low frequency calibration

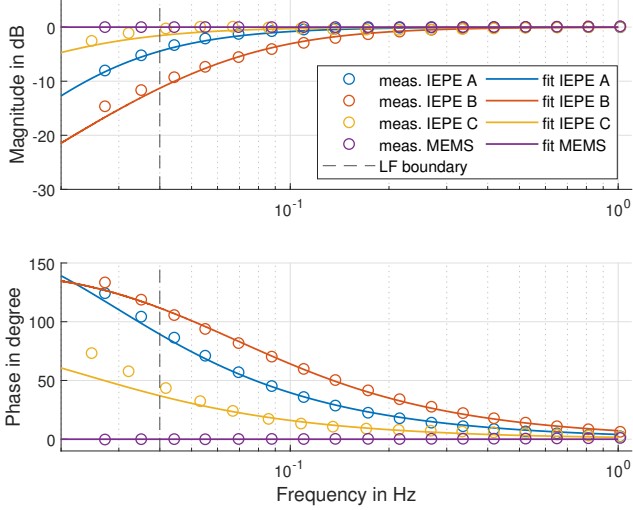

**Figure 12.** Results of the sensor calibration in comparison to the fitted transfer function with the low frequency (LF) boundary for the optimisation.

A filter model is required to apply the calibration of IEPE sensors to measurement data. To identify a filter model using the method proposed in Section 2.5, the number of cascaded high-passes and the gain of the shelf has to be defined in advance. The number of cascaded high-passes is determined by analysing of the high-pass behaviour of the corresponding sensor. A

first-order high-pass filter would lead to a phase shift of $90°$ within three decades in the low-frequency range, as shown in Figure 4. Steeper phase response correspond to higher order high-pass filter. This consideration results in a filter order of three for the sensor IEPE A, whereas the sensor IEPE B can be modelled from two cascaded high-passes and IEPE C with a first-





order high-pass. In addition to the high-pass filters of the sensors, a further high-pass is introduced for the signal conditioner. When setting the shelf gain $G_{\text{shelf}}$, it should be taken into account that a higher gain reduces the settling time of the filter and
leads to higher noise suppression in the low-frequency range. However, in addition to an amplitude deviation, the shelf gain has a big influence on the phase response of the model as shown in Figure 4. To prevent this phase error of the shelf from distorting the filter model, only measurement data above $0.04\,\text{Hz}$ are used for the filter parameter identification. The adaptation of the model to the determined transfer functions is achieved using the complex objective function as shown in Equation 20 and using the Global Pattern Search algorithm. The transfer functions of the filter models are shown superimposed with the
measured transfer behaviour in Figure 12. Generally, the identified models fit well with the measured data. The phase deviation in the lower frequency range is caused by the shelf filter. Additionally small amplitude deviations can be observed. IEPE C in particular seems to have a more complex transfer behaviour than a first-order high-pass filter. Table 2 lists the selected orders, shelf amplitudes and cutoff frequencies for each sensor.

**Table 2.** Filter models of the investigated IEPE sensors

|  | IEPE A | IEPE B | IEPE C |
|---|---|---|---|
| order sensor | 3 | 2 | 1 |
| $G_{\text{shelf}}$ in dB | -60 | -60 | -30 |
| $f_c$ sensor in Hz | 3x0.0251 | 2x0.0651 | 0.0311 |
| $f_c$ supply in Hz | 0.0106 | 0.0106 | 0.0106 |

## 5    Application

In this section, we apply the calibration results to measurement data. First, the noise of the calibrated sensors is analysed and the sensor types are compared. Second, the calibration filters are applied to time series measurement data.

### 5.1    Noise level of the measurement chain

The noise level of the measurement chain is an important parameter for sensor selection. In the low-frequency range, the 1/f noise typically dominates the noise amplitude. Furthermore, the acceleration amplitudes in the low-frequency range are
often low and thus the noise level is more important for low-frequency measurements than for high-frequency measurements. In addition, the calibration of the IEPE sensors using an inverse high-pass filter increases the noise amplitude in the lower frequency range. We determine the noise level by means of a coherence analysis of two sensors of the same type as described in Section 2.2. The spectral noise is calculated according Equation 5 using data measured during a time span of 100 minutes. For the evaluation with the Welch method, a rectangular window of $1000\,\text{s}$ length is used. This leads to a frequency resolution
of $0.001\,\text{Hz}$. The resulting spectral noise of the three sensor types is shown in Figure 13.



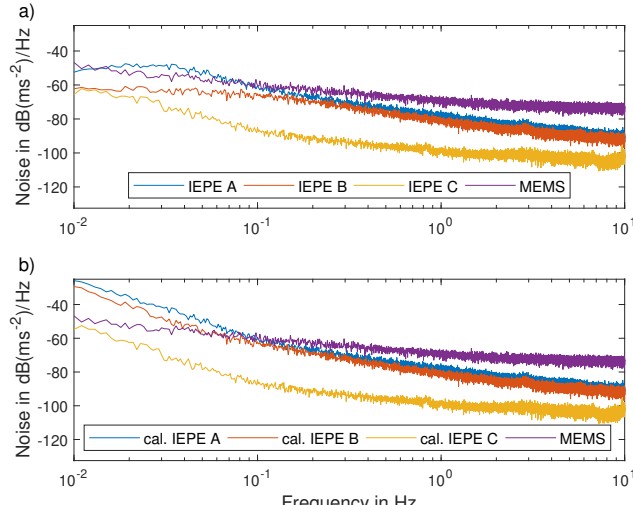

**Figure 13.** Noise level of each sensor; a) without the low frequency calibration; b) with the applied calibration filter

In panel a) of Figure 13, the uncalibrated sensor IEPE C shows a typical 1/f noise characteristic. In case of the sensors IEPE A and B, the noise flattens below $0.04\,\text{Hz}$ due to the high-pass characteristic of the sensor and the signal conditioner. The 1/f characteristic is recovered for all IEPE sensors, when the data is calibrated, as shown in panel b) of Figure 13.

An interesting effect is that IEPE B has significantly lower noise in the low-frequency range than IEPE A in the uncalibrated
data. However, after the calibration is applied, the differences in the noise level between IEPE A and B diminish. Therefore, only calibrated sensor signals should be used when comparing sensors based on their noise level. The noise level of the MEMS sensor is only better than that of the calibrated sensors IEPE A and B below $0.1\,\text{Hz}$. Thus, using these IEPE sensors, a SNR higher than with the MEMS sensor can be expected above $0.1\,\text{Hz}$. The seismic sensor IEPE C has a significantly better noise performance than the other sensors down to $0.01\,\text{Hz}$. However, it also has a smaller measuring range which makes it suitable
only for applications with low acceleration amplitudes.

## 5.2  Calibration of measurement data

To calibrate measurement data, the developed filter model is inverted and applied to the time domain data. In the model, a shelf is added to the filter model to limit the amplitude increase at low frequencies. However, the shelf leads to a phase error. For a phase-true calibration, the shelf gain cannot be set high enough to suppress the low-frequency noise and keep the settling
time low. To avoid amplifying the low-frequency noise too much, frequency components below the frequency range of interest should be removed after the calibration. However, this also changes the phase response when used in a real-time filtering scenario. If the calibration is applied in a post-process, the phase can be maintained by filtering forwards and backwards in time.





To determine the required high-pass filters, the objective function from Equation 21 is used. For this purpose, it is necessary

to determine the maximum gain. The maximum gain $G_{max}$ is chosen to be as low as possible without affecting the amplitude response above $0.05\,\text{Hz}$. Figure 14 shows the resulting calibration filters. The settings of the high-pass filters applied for the removal of low frequency noise are listed in Table 3.

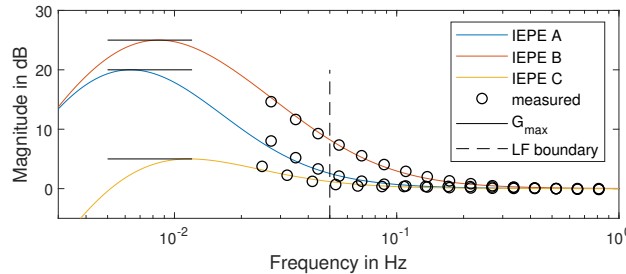

**Figure 14.** Magnitude of the inverse filter model of the sensors with the additional high-pass filter

**Table 3.** Settings high-pass filters of the calibration

|  | IEPE A | IEPE B | IEPE C |
|---|---|---|---|
| order | 2 | 2 | 2 |
| $G_{max}$ (dB) | 20 | 25 | 5 |
| $f_c$ in Hz | 0.0081 | 0.0077 | 0.0073 |

In general, the lower the cut-off frequency and order of the high-pass of the sensor, the lower the maximum gain can be set. A lower maximum gain thus leads to an increased noise suppression below $0.05\,\text{Hz}$.

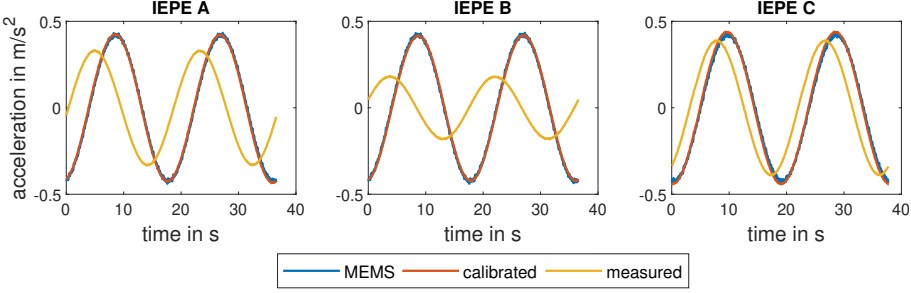

**Figure 15.** Time histories of calibrated and uncalibrated IEPE sensors compared to MEMS sensors with the excitation frequency f=0.055 Hz

We apply the calibration procedure to measurement data taken during another run of the centrifuge. The graphs in Figure 15 show two periods of rotation at a frequency of $0.055\,\text{Hz}$. For comparison, the signal of the MEMS sensor is shown. For these plots, an additional second order low-pass filter with a cut-off frequency of $2.5\,\text{Hz}$ is applied forwards and backwards in time to remove high-frequency signal contamination. Especially the sensors IEPE A and B have a significant phase and amplitude





deviation at this excitation frequency. The deviations are not as pronounced in the case of IEPE C due to its excellent low-
frequency performance. Using the filter models identified for the respective IEPE sensors, the amplitude and phase deviation
between the MEMS and the IEPE sensors can be completely corrected.

## 5.3 Calibration of tower vibrations of a wind turbine

The MEMS sensor, as well as the sensors IEPE B and C combined with the custom IEPE conditioner were used to measure the
tower vibrations of a $3.4\,\mathrm{MW}$ on-shore wind turbine. The sensor setup is shown in Figure 16. This measurement data enables

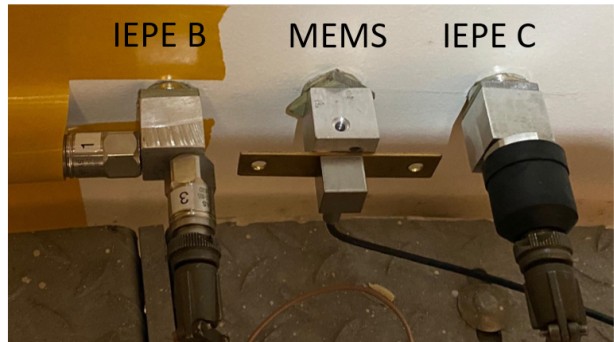

**Figure 16.** Measurement setup of the sensor comparison in the tower of a wind turbine.

a sensor comparison in a realistic scenario. The measuring point is located at a height of $96\,\mathrm{m}$. During the startup process of
the wind energy turbine, strong tower vibrations are observed in the measurement data. Figure 17 shows a part of this vibration

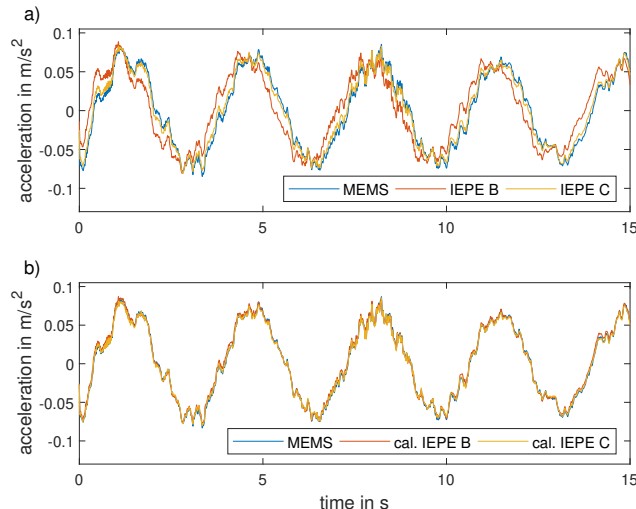

**Figure 17.** Acceleration vs. time of uncalibrated a) and calibrated b) IEPE sensors compared to MEMS sensors in a wind turbine tower
during a startup process.



time series. The fundamental oscillation frequency in this data is around $0.3\,\text{Hz}$. For better visualisation, all signal components above $10\,\text{Hz}$ are removed by a low-pass filter. Panel a) shows the time series of the uncalibrated sensor data. In this case, a phase shift is discernible between the sensors. This phase shift is larger from IEPE B to the MEMS sensor when compared to IEPE C. This observation is in agreement with the previously determined transfer characteristics of the sensors. By applying the calibration filters, the phase of the signal can be corrected, as shown in Panel b).

During operation of the wind turbine, lower frequency signal components can be observed. By applying a double time integration, these signal components become visible in the measurement signal. The displacement estimation is shown in Figure 18. To prevent drift due to integration, the measurement data below $0.04\,\text{Hz}$ are removed by a high-pass filter. Panel a) shows the calculated displacement of the uncalibrated IEPE sensors compared to the MEMS sensor. Besides the phase error of the IEPE sensors, an amplitude error is visible. This leads to a different time series. The calibration filters can correct both errors. This is shown in Panel b). It should be noted that a tilt error occurs in the measurement data due to gravitational acceleration. (Tarpø et al., 2021). This is not taken into account in the evaluation.

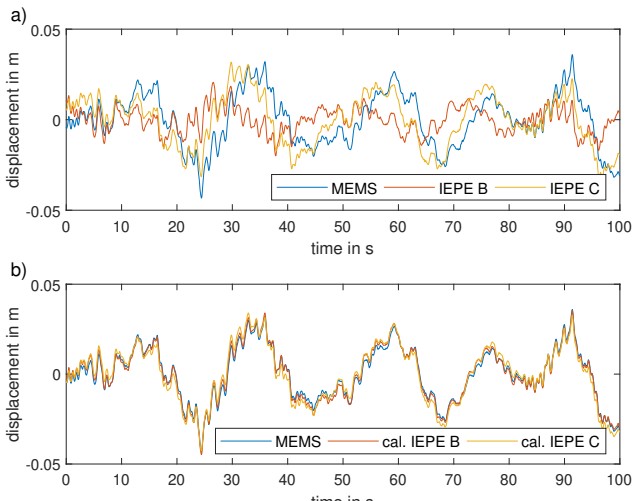

**Figure 18.** Displacement vs. time of uncalibrated a) and calibrated b) IEPE sensors compared to a MEMS sensor in a wind turbine tower during operation.

The advantage of the lower noise level of IEPE sensors in frequency ranges above $0.1\,\text{Hz}$ becomes apparent at a low signal level. This is expected at lower measurement planes in the tower as well as during downtime of the wind turbine. Of the latter an auto power spectral density (PSD) is shown in Figure 19. In this case, the lower noise level of the IEPE C is observable in comparison with the other sensors. Above $0.02\,\text{Hz}$, the IEPE B sensor shows better noise level than the MEMS Sensor. These values are for illustrative purposes only, as they depend on the measured acceleration and thus on the excitation of the structure.





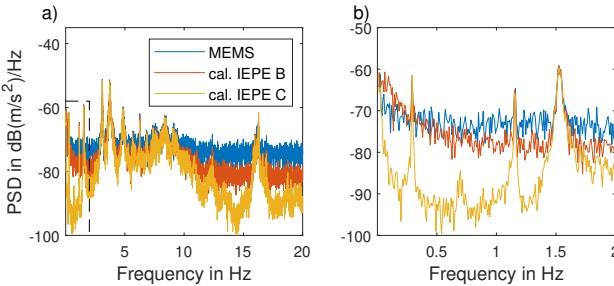

**Figure 19.** Auto power spectral density (PSD) of the sensors during downtime of the wind turbine in the frequency range a) 0-25 Hz and b) 0-2 Hz.

## 6 Summary and outlook

In this work, we demonstrate that measurements of very low-frequency structural dynamics down to $0.05\,\mathrm{Hz}$ can be achieved using IEPE accelerometers. To this end, we introduce a custom IEPE signal conditioner with low noise and a low cutoff frequency. The necessary low frequency calibration of the measurement chain is carried out in the range from $0.027\,\mathrm{Hz}$ up to $1\,\mathrm{Hz}$ using a tilted plane centrifuge. For the signal calibration, it is sufficient to model the transfer behaviour of IEPE sensors including the signal conditioner with a high order high-pass filter. In comparison to a MEMS sensor, the investigated calibrated

IEPE sensors have a better signal to noise ratio in the range above $0.1\,\mathrm{Hz}$. The sensor comparison in a tower of a wind turbine shows that the calibrated IEPE sensors provide amplitude and phase-confident signals above $0.05\,\mathrm{Hz}$. The lower noise level of the IEPE sensors leads to improved measurements at low acceleration amplitudes, such as downtime of the wind turbine, low wind speed and lower measurement planes. Precise measurements, also for low amplitudes, are an important prerequisite for life time extrapolation based on measurements.

In the future, a precise uncertainty investigation and refinement of the presented calibration method should be carried out. Therefore, a more accurate phase sensor and a centrifuge optimised for low frequencies should be used. In addition, more precise filter models for modelling the IEPE sensors should be investigated. The techniques developed in this paper can be used to estimate the displacement and strain of large structures using low-noise IEPE accelerometers. Thus, further investigations should validate the displacement estimated from the acceleration using low-noise IEPE accelerometers by comparison to

independent displacement measurements. In addition, a modal expansion technique can be used to estimate strains in areas of structures where measurement is difficult or impossible, such as offshore structures below sea level. For this purpose, the influence and the correction of the tilt error of accelerometers in the monitoring of support structures of wind turbines should be investigated.

In many applications of SHM, heterogeneous sensor networks are applied. In contrast, operational modal analysis techniques

rely on a homogeneous sensor network to obtain in-phase mode shapes. In frequency ranges with linear transfer behaviour of all sensors, the use of heterogeneous sensor networks is possible for modal analysis. The calibration method for IEPE accelerometer allows them to be included in the heterogeneous sensor networks in the low frequency range without phase and



amplitude errors as well. Furthermore, the influence of the lower noise level of the measurement chain on the uncertainty of the modal analysis should be investigated.

*Data availability.*  The data that support the findings of this study are available from the corresponding author, CJ, upon request.

*Author contributions.*  BH devised the original idea of this research. CJ and BH designed the experiment. CJ was responsible for carrying out the experiments and the analysis of the measurements. TG and RR supervised the work. CJ wrote the manuscript. All authors reviewed the manuscript.

*Competing interests.*  RR is associate editor of WES.

*Acknowledgements.*  We greatly acknowledge the financial support of the Federal Ministry for Economic Affairs and Energy of Germany (research projects *Deutsche Forschungsplattform für Windenergie*, FKZ 0325936E and *PreciWind-Präzises Messsystem zur berührungslosen Erfassung und Analyse des dynamischen Strömungsverhaltens von WEA-Rotorblättern*, FKZ 03EE3013B) that enabled this work. In addition, we are grateful to the *Deutsche WindGuard GmbH*, as well as the *Bremer Institut für Messtechnik, Automatisierung und Qualitätswissenschaft* (BIMAQ) for their support during the measurement campaign."



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
