# Peer review of "Very low-frequency IEPE accelerometer calibration and application to a wind energy structure"

_Wind Energy Science, 2021_

## Author Comment (AC1)

**Very low-frequency IEPE accelerometer calibration and application to a wind energy structure**

**Answers to the Reviewer's comments**

The authors would like to thank the Editor and Reviewers for their time and effort to review the article and their constructive comments. We appreciate the chance to clarify the commented points. This will undoubtedly help in improving the quality of the manuscript. The article is revised according to the Reviewers' remarks and queries, with detailed responses and explanations of the edits given below. All edits in the manuscript have been highlighted in red color.

We note that during the revision, the amount and numberings of lines have been changed. Therefore, in the proceeding paragraphs, we refer to the revised article.

**Reviewer 1**

**1) One piece needing more discussion is IEPE simulator, and its design, parameter determination, and potential impacts on the overall calibration approach will be interesting to the audience.**

Thank you for the comment. We use an commercial IEPE simulator (PCB-401B04). Therefore we do not have a wiring diagram of the simulator. To express this, we have added the word 'commercial' to the text in line 266:

*[...] We employ a commercial IEPE simulator with a flat low-frequency response down to DC. [...]*

The main purpose of the IEPE simulator calibration is to explain the influence of different signal conditioners. In addition, the simulator was used to check the functionality of the customised signal conditioner. Since in section 4 this entire measurement chain is calibrated including the supply, the IEPE simulator does not influence the calibration results in any way. Therefore the following paragraph was added at the end of section 3 in Line 293:

*This calibration method covers the entire measurement chain including the signal conditioner. Therefore, the calibration of the signal conditioner with the IEPE simulator is not absolutely necessary for applications. The dedicated calibration of the signal conditioner may still be useful to check its functionality. Further, it illustrates the influence of the signal conditioner on the low-frequency performance of the measurement chain and also enables a validity check of filter coefficients determined in the calibration using the centrifuge.*

**2) However, the approach appears needing a lot of customization and hard to have huge impact on wind or sensing industry. On the other hand, the contents appear more appropriate to be released through a publication venue with more sensing elements than wind.**

In fact, the use of IEPE accelerometers in the low frequency range requires more background knowledge than the use of MEMS sensors. However, more and more IEPE accelerometers are being developed for the low frequency range, especially for the wind energy sector, to take advantage their lower noise level. Therefore, in order to obtain measurement data that matches to physical quantities, knowledge of the transfer behaviour is essential. On the one hand, this article is intended to show a possibility how calibration can be carried out in a relatively inexpensive way in the low frequency range and how filter models can be used to calibrate the measurement data. On the other hand, it is intended to point out possible pitfalls to researchers and industry when using IEPE sensors in the low-frequency range. Therefore, we have deliberately chosen an application-related wind energy journal for this topic.

In order to make the reference to the wind energy sector more clear, we have expanded the state of the art with regard to wind turbines on the basis of these comments and the comments of the second reviewer.

In Line 23 we add

*However, field experiences in offshore wind energy turbines show that strain sensors are less reliable than accelerometers for long-term applications (Maes et al., 2016).*

In Line 27 we add

*Acceleration sensors are commonly used in the wind energy industry for support structure monitoring of wind turbines. In the low frequency range, DC-coupled Micro-electro-mechanical systems (MEMS) capacitive accelerometers are usually applied, because these sensors have a linear transfer behaviour in the low frequency range (Anslow and O'Sullivan, 2020).*

and in Line 40

*Occasionally, IEPE sensors are also used for monitoring the tower of offshore wind turbines in the frequency range above $0.05\,\mathrm{Hz}$ due to their low noise level (Weijtjens et al., 2017). However, to the best of our knowledge, the transfer behaviour of IEPE sensors in the low-frequency range has not been considered specifically so far. For laboratory experiments on rotor blades, there are experiments with IEPE acceleration sensors where a calibration for frequencies starting at $0.5\,\mathrm{Hz}$ was carried out (Gundlach and Govers, 2019).*

Therefore, we have included three new references:

Anslow,R. and O'Sullivan, D., Choosing the Best Vibration Sensor for Wind Turbine Condition Monitoring, in: Analog Dialogue, vol.54, https://www.analog.com/media/en/analog-dialogue/volume-54/number-3/choosing-the-best-vibration-sensor-for-wind-turbine-condition-monitoring.pdf,2020

Gundlach, J. and Govers, Y.: Experimental modal analysis of aeroelastic tailored rotor blades in different boundary conditions, in: Journal ofPhysics: Conference Series, vol. 1356, p. 012023, IOP Publishing, 2019

Weijtjens, W., Verbelen, T., Capello, E., and Devriendt, C.: Vibration based structural health monitoring of the substructures of five offshore wind turbines, Procedia Engineering, 199, 2294–2299, https://doi.org/10.1016/j.proeng.2017.09.187, 2017

**Reviewer 2**

**1) In introduction section please extend the literature review about the sensors used today in the wind industry sector**

Thank you for pointing out this deficiency. We have clarified the reference to wind energy and added more specific literature sources. In Line 23 we add

*However, field experiences in offshore wind energy turbines show that strain sensors are less reliable than accelerometers for long-term applications (Maes et al., 2016).*

In Line 27 we add

*Acceleration sensors are commonly used in the wind energy industry for support structure monitoring of wind turbines. In the low frequency range, DC-coupled Micro-electro-mechanical systems (MEMS) capacitive accelerometers are usually applied, because these sensors have a linear transfer behaviour in the low frequency range (Anslow and O'Sullivan, 2020).*

and in Line 40

*Occasionally, IEPE sensors are also used for monitoring the tower of offshore wind turbines in the frequency range above $0.05\,\mathrm{Hz}$ due to their low noise level (Weijtjens et al., 2017). However, to the best of our knowledge, the transfer behaviour of IEPE sensors in the low-frequency range has not been considered specifically so far. For laboratory experiments on rotor blades, there are experiments with IEPE acceleration sensors where a calibration for frequencies starting at $0.5\,\mathrm{Hz}$ was carried out (Gundlach and Govers, 2019).*

Therefore, we have included three new references:

Anslow,R. and O'Sullivan, D., Choosing the Best Vibration Sensor for Wind Turbine Condition Monitoring, in: Analog Dialogue, vol.54, https://www.analog.com/media/en/analog-dialogue/volume-54/number-3/choosing-the-best-vibration-sensor-for-wind-turbine-condition-monitoring.pdf,2020

Gundlach, J. and Govers, Y.: Experimental modal analysis of aeroelastic tailored rotor blades in different boundary conditions, in: Journal ofPhysics: Conference Series, vol. 1356, p. 012023, IOP Publishing, 2019

Weijtjens, W., Verbelen, T., Capello, E., and Devriendt, C.: Vibration based structural health monitoring of the substructures of five offshore wind turbines, Procedia Engineering, 199, 2294–2299, https://doi.org/10.1016/j.proeng.2017.09.187, 2017

**2) Fig. 13: please elaborate more on the physics or reasons behind the differences seen in this figure**

This objection is justified. We have included an explanation for the different noise levels in line 387:

*Differences in the design of the internal electronic components of the sensor types used in this study lead to varying noise levels. MEMS acceleration sensors are capacitive sensors that require a carrier frequency for measurement, leading to a higher noise level. The different noise levels exhibited by the IEPE sensors have several reasons. A*

*sensor with a lower measuring range usually houses a larger piezoelectric crystal, which in turn leads to a lower noise level due to lower impedance. Another influence is the integrated electronic pre-amplifier, which significantly affects the noise level of the IEPE sensor (Levinzon, 2005).*

**3) It would be nice if the authors can provide a table in section 6 providing suggestions for selecting suitable sensors for different range of frequencies in different wind turbine components**

It is difficult to make a general statement about specific sensors and application ares, as there may be other application restrictions, such as whether the wave frequency spectrum should be captured, the noise level of the measurement system used, etc. We have therefore tried to make a statement as general as possible regarding the frequency ranges and inserted it in line 447:

*Taking into account the transfer behaviour of the measurement chain, the use of IEPE accelerometers designed for the low-frequency range is therefore recommended for all wind turbine components in a frequency range above 0.1 Hz. In the range of 0.05 Hz and 0.1 Hz, both sensor types have similar performance and a decision has to be made considering the requirements in each particular case. This recommendation is only valid when signal conditioners with a very low high-pass cut-off frequency are employed. Seismic IEPE sensors should not be considered for rotating systems due to their low measuring range. Regardless of the acceleration sensor type, the tilt error has to be considered for measurements in the low-frequency range due to the contamination of the structural acceleration with the gravitational acceleration caused by the bending of the structure (Tarpø et al., 2021). This should be compensated for when exact acceleration amplitudes are desired.*